# Evaluating the Ability of the Pre-Launch TanSat-2 Satellite to Quantify Urban CO$_2$ Emissions

**Kai Wu** [1] , **Dongxu Yang** [1,2,*] , **Yi Liu** [1,2,3] , **Zhaonan Cai** [1,2] , **Minqiang Zhou** [1,2] , **Liang Feng** [4,5] **and Paul I. Palmer** [4,5]

1    Carbon Neutrality Research Center, Institute of Atmospheric Physics, Chinese Academy of Sciences, Beijing 100029, China

2    Key Laboratory of Middle Atmosphere and Global Environment Observation, Institute of Atmospheric Physics, Chinese Academy of Sciences, Beijing 100029, China

3    University of Chinese Academy of Sciences, Beijing 100049, China

4    School of GeoSciences, University of Edinburgh, Edinburgh EH9 3FF, UK

5    National Centre for Earth Observation, University of Edinburgh, Edinburgh EH9 3FF, UK

*    Correspondence: yangdx@mail.iap.ac.cn

**Abstract:** TanSat-2, the next-generation Chinese greenhouse gas monitoring satellite for measuring carbon dioxide (CO$_2$), has a new city-scale observing mode. We assess the theoretical capability of TanSat-2 to quantify integrated urban CO$_2$ emissions over the cities of Beijing, Jinan, Los Angeles, and Paris. A high-resolution emission inventory and a column-averaged CO$_2$ (X$_{CO_2}$) transport model are used to build an urban CO$_2$ inversion system. We design a series of numerical experiments describing this observing system to evaluate the impacts of sampling patterns and X$_{CO_2}$ measurement errors on inferring urban CO$_2$ emissions. We find that the correction in systematic and random flux errors is correlated with the signal-to-noise ratio of satellite measurements. The reduction in systematic flux errors for the four cities are sizable, but are subject to unbiased satellite sampling and favorable meteorological conditions (i.e., less cloud cover and lower wind speed). The corresponding correction to the random flux error is 19–28%. Even though clear-sky satellite data from TanSat-2 have the potential to reduce flux errors for cities with high CO$_2$ emissions, quantifying urban emissions by satellite-based measurements is subject to additional limitations and uncertainties.

**Keywords:** TanSat-2 satellite mission; urban CO$_2$ emissions; atmospheric CO$_2$ inversion; OSSE





## 1. Introduction

Carbon dioxide (CO$_2$) is one of the most important greenhouse gases, and was responsible for about 80% of global CO$_2$ equivalents in 2022 (https://gml.noaa.gov/aggi/ (accessed on 25 September 2023)) [1]. Quantifying urban CO$_2$ emissions has become a focus of climate change mitigation efforts to limit global warming and achieve carbon neutrality [2–6]. Satellite-based measurements of atmospheric CO$_2$ aim to quantify surface CO$_2$ fluxes [7]. Here, we explore the theoretical potential of the Chinese next-generation greenhouse gas monitoring satellite (TanSat-2) to infer urban CO$_2$ emissions.

Anthropogenic CO$_2$ emissions have been quantified using data from ground-based instruments [8–17] to low-Earth-orbiting satellites such as GOSAT [18–21] and OCO-2 [22–25], either directly as X$_{CO_2}$, the atmospheric column-averaged dry-air mole fraction of CO$_2$ [26–28], or by monitoring tropospheric NO$_2$ (a short-lived trace gas related to the combustion of fossil fuels) [29–35]. The main advantage of low-Earth-orbiting satellites is their global coverage, which is subject to cloud cover and aerosol loading [36,37]. The challenge of using data from existing satellites to quantify urban CO$_2$ emissions is that the probability of sampling clear skies over individual cities is low due to the relatively long revisit time (3–16 days) and small footprints of the instruments. While this limited capability is shared with GOSAT-2 [38] and TanSat [39–42], it is expected to be improved in the near future with the France–UK MicroCarb satellite [43–45], which has a city-observing model, and with the

Copernicus $CO_2$ Monitoring Mission [46,47]. These missions should dramatically increase the density of relevant data.

Satellite-based $X_{CO_2}$ measurements are being developed at finer sub-city scales to constrain urban $CO_2$ emissions with high accuracy and precision. Kiel et al. (2021) [48] estimated urban $X_{CO_2}$ enhancements in Los Angeles ranging from 0 to 6 ppm using data collected by the NASA Orbiting Carbon Observatory-3 (OCO-3, installed on the International Space Station in 2019) Snapshot Area Maps (SAMs) observing mode [49,50]. They found that high-density satellite measurements with sufficient accuracy have the potential to detect changes in anthropogenic $CO_2$ emissions over cities. The $X_{CO_2}$ data collected by the NASA Orbiting Carbon Observatory-2 satellite (OCO-2, in orbit since 2014) [7,24] have been used to constrain $CO_2$ emissions in urban areas [26,28]. Lei et al. (2021) [28] examined OCO-2 data availability and suggested collecting high-frequency data near metropolitan areas to better constrain the trend of urban $CO_2$ emissions. Recent studies have empirically related city-scale $CO_2$ emission estimates to urban population density [51,52].

The launch of TanSat-2 is planned for 2025. TanSat-2 is a satellite cluster consisting of two to three satellites measuring $X_{CO_2}$ at an across-track swath of 2900 km and with a pixel size of 2 km $\times$ 2 km. The precision of its $X_{CO_2}$ measurements is expected to be less than 1 ppm. It is intended to verify satellite data using ground-based measurements from the Total Carbon Column Observing Network [53–55] and EM27/SUN measurements [56] for optimizing parameters in satellite sampling. In this paper, we evaluate the theoretical ability of TanSat-2 to detect urban $CO_2$ emission signatures based on a closed-loop inversion system (Figure 1). We simulate synthetic data sampled by TanSat-2 in Beijing (BJ), Jinan (JN), Los Angeles (LA), and Paris (PR) while accounting for the impacts of cloud cover and aerosol loading. We compare the effectiveness of correcting flux bias and random errors in different cities. Finally, we discuss limitations and uncertainties in linking these results to real-data inversion experiments.

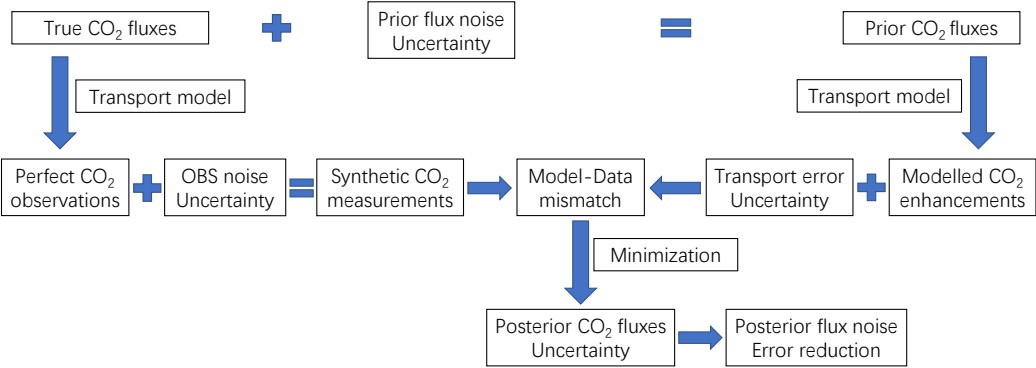

**Figure 1.** Flow chart of the Observing System Simulation Experiment (OSSE) for urban $CO_2$ inversion.

## 2. Data and Methods

### 2.1. TanSat-2 Configuration

The next-generation Chinese greenhouse gas monitoring satellite known as TanSat-2 is the continuation mission of China's first carbon monitoring satellite, TanSat. TanSat-2 will fly at a medium-Earth orbit with an apogee of 7840 km and a perigee of 522 km (Figure 2). Previous studies indicate that more than 80% of anthropogenic $CO_2$ emissions are concentrated in the region of 15°–55°N, which the apogee of TanSat-2 is placed over in order to measure most fossil fuel $CO_2$ emissions. Three bands, $O_2$-A (0.747–0.773 μm), $CO_2$ and $CH_4$ (1.590–1.675 μm), and $CO_2$ (1.990–2.095 μm), provide continuous measurements of the atmospheric absorption spectrum of $CO_2$ and $CH_4$, with an expected retrieval precision of less than 1 ppm and 8 ppb to account for clouds and aerosol loading. In addition, there is a visible band for measuring $NO_2$ to identify and separate anthropogenic emissions from natural carbon fluxes. Overall, TanSat-2 includes normal push broom, target, and glint modes. A new cloud and aerosol polarization imager will be installed as

well, providing additional information to reduce errors due to clouds and aerosol particles.

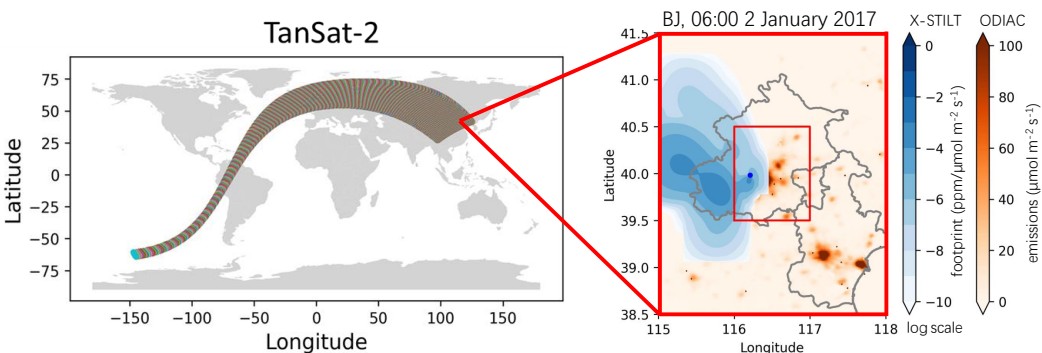

**Figure 2.** TanSat-2 sampling trajectory and the simulated footprint of a cloud-free sample at 06:00 (UTC) 2 January 2017 over Beijing (BJ) with monthly mean ODIAC emissions. Footprint values are plotted on a logarithmic scale.

We simulated synthetic satellite measurements over BJ, JN, LA, and PR for four arbitrary clear-sky days in January and April 2017 (Figure 3). Following the same method described in Wu et al. (2023) [45], we used ERA5 total cloud cover reanalysis data at $0.25° \times 0.25°$ resolution [57] to screen out samples contaminated by clouds while accounting for the organization of cloud distribution and the randomness of the impact of clouds and aerosol particles on satellite measurements. Of the more than one thousand individual samples over each city, our method identified 397 (BJ), 255 (JN), 217 (LA), and 307 (PR) samples as cloud-free in each city (Figure 3).

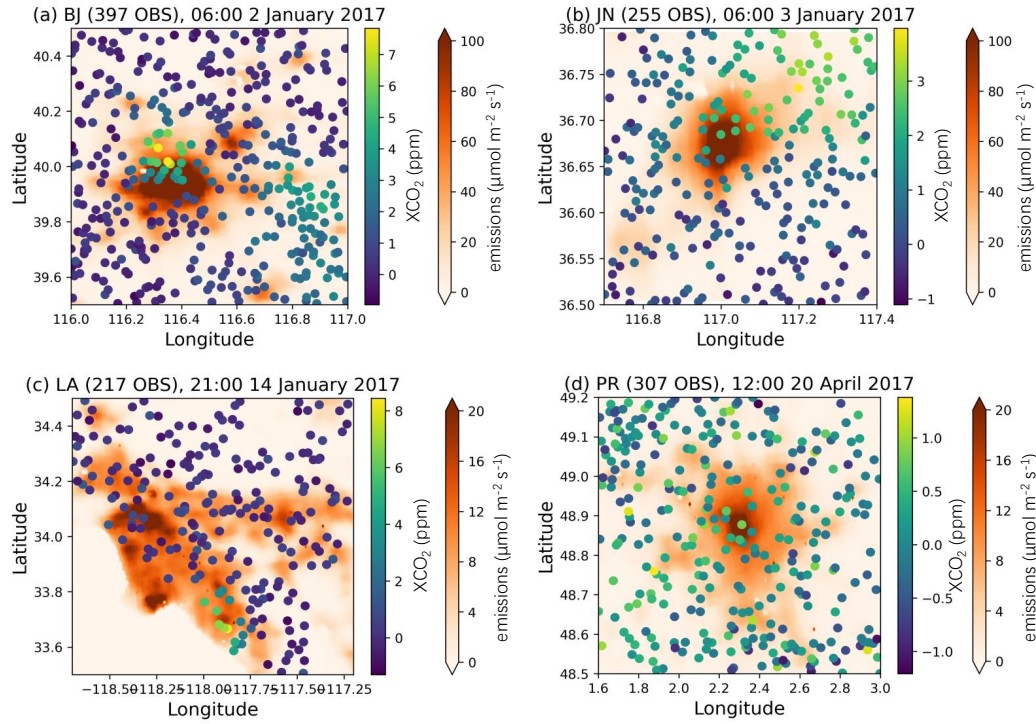

**Figure 3.** Monthly mean ODIAC emissions and synthetic cloud-free $CO_2$ samples over Beijing (**a**), Jinan (**b**), Los Angeles (**c**), and Paris (**d**).

## 2.2. Anthropogenic $CO_2$ Emission Inventory

We used the Open-source Data Inventory for Anthropogenic $CO_2$ (ODIAC, version 2020b) for monthly mean $CO_2$ emissions from fossil fuels at a spatial resolution of 1 km × 1 km [58,59]. This data product uses satellite observations of night-time light and

power plant profiles, including emission intensity and geographic location, to distribute $CO_2$ emission estimates from fossil fuel combustion at the country level. Sources of systematic and random flux errors are due to spatial and temporal disaggregation that vary by emission sector at different scales (from national to urban) and uncertainties in the use of night-time light data as an emission proxy. We used ODIAC to define our broader study domains for urban areas and to ensure that anthropogenic source regions that lie outside the spatial extent of TanSat-2 were included (Figure 3). BJ (9.84 t$CO_2$ s$^{-1}$) and JN (1.63 t$CO_2$ s$^{-1}$) emissions of $CO_2$ are concentrated in the city centre, with high emissions beyond 60 μmol m$^{-2}$ s$^{-1}$ (Figure 3), home to about 21.54 and 8.7 million people. LA (3.67 t$CO_2$ s$^{-1}$) and PR (1.81 t$CO_2$ s$^{-1}$) show fewer $CO_2$ emissions ranging from 10 to 20 μmol m$^{-2}$ s$^{-1}$ (Figure 3), home to about 3.85 and 2.16 million people, where emissions are distributed with the expansion of city.

### 2.3. Atmospheric Transport Model

We used the column version of the Stochastic Time-Inverted Lagrangian Transport model (X-STILT) [60–62] to link surface $CO_2$ fluxes to variations in atmospheric column $CO_2$ at the locations of satellite sampling. X-STILT tracks the movement of air parcels backwards in time for 24 h. We applied a typical averaging kernel profile of OCO-2 and pressure weighting functions to the model fields to describe the footprints of satellite-based atmospheric $CO_2$ measurements (Figure 2). The footprints describe the sensitivity of $CO_2$ columns at the receptors (locations where the satellite observes the atmosphere) to upwind surface fluxes. In order to drive air parcels in X-STILT, we used meteorological data from the Global Forecast System with a horizontal resolution of 0.25 degrees (GFS0.25, https://www.ready.noaa.gov/data/archives/gfs0p25/ (accessed on 25 September 2023)) [63]. A total of 3000 air parcels evenly distributed from the surface to a 3 km height were released from the atmospheric column of each observation. We simulated footprints for the cloud-free observations sampled by TanSat-2. The sum of the convolution of the footprints and the ODIAC inventory represents the urban $CO_2$ enhancements from upwind $CO_2$ fluxes, as sampled by air parcels arriving at the locations of each sample.

### 2.4. Urban $CO_2$ Inversion System

We followed the same method described in Wu et al. (2023) [45] to configure an urban $CO_2$ inversion system with synthetic satellite measurements. Figure 1 describes the experimental design we followed to assess the theoretical ability of TanSat-2 sampling to quantify urban emissions of $CO_2$. We used the ODIAC emissions as the true state. The corresponding $CO_2$ column enhancements were generated from the true fluxes using the X-STILT transport model. We added synthetic observation noise to each cloud-free scene based on simulations of cloud cover and aerosol loading. The random measurement errors ranged between −2 ppm to 2 ppm, with a standard deviation of 0.41 ppm to 0.54 ppm. We added an unbiased 20% random error to account for atmospheric transport errors [64,65]. Later in this paper, we examine the impacts of different transport errors (described by the observation error covariance matrix) on the error reduction of flux inversion.

Evaluating the ability to reduce a priori flux errors (including systematic and random errors) was the primary objective of this study. We assumed respective mean systematic and random flux error of 2 μmol m$^{-2}$ s$^{-1}$ for the prior state [66,67]. The total systematic and random flux errors are constrained by the Chi-square test, which should be close to one (meaning that the posterior state is weighted by balancing the information from the data and the prior state) [16]. We used an eigenvalue decomposition method to generate a priori flux noise from the flux error covariance matrix, which is spatially correlated with an exponentially decaying function of the distance between emission grids. The spatial correlation length was assumed to be 10 km [68,69]. The vector of prior flux noise was calculated by multiplying the eigenvector of the flux error covariance matrix with a normal distribution vector characterizing the systematic and random flux errors.

We used the Maximum A Posteriori (MAP) inverse method [70–72], in which we solved for a posteriori $CO_2$ emissions by minimizing a cost function [67] that describes the mismatch between the model-calculated enhancements and the measurements while accounting for a priori and measurement uncertainties. Minimizing the cost function results in the following expressions:

$$\hat{x} = x_0 + (\mathbf{HB})^{\mathrm{T}}(\mathbf{HBH}^{\mathrm{T}} + \mathbf{R})^{-1}(y - \mathbf{H}x_0), \tag{1}$$

$$\hat{\mathbf{S}} = \mathbf{B} - (\mathbf{HB})^{\mathrm{T}}(\mathbf{HBH}^{\mathrm{T}} + \mathbf{R})^{-1}(\mathbf{HB}), \tag{2}$$

where $\hat{x}$ and $\hat{\mathbf{S}}$ denote the a posteriori state of grid-based $CO_2$ emissions and the associated error covariance matrix, $x_0$ and $\mathbf{B}$ denote the a priori emissions and the associated error covariance matrix, the measurement vector $y$ includes the atmospheric $CO_2$ column enhancements (with the associated errors described by the observation error covariance matrix $\mathbf{R}$, including measurement errors and atmospheric transport errors), and $\mathbf{H}$ denotes the Jacobian matrix that describes the sensitivity of $CO_2$ column enhancements to changes in surface $CO_2$ emissions.

To evaluate the theoretical performance of TanSat-2 sampling on improving a priori knowledge of urban $CO_2$ emissions, we used an error reduction metric ($\eta$) that takes into account differences between the a priori and a posteriori random flux errors [73]:

$$\eta = \left[ 1 - \left( \frac{\hat{\mathbf{S}}_{i,i}}{\mathbf{B}_{i,i}} \right)^{1/2} \right] \times 100\%, \tag{3}$$

where the subscripts denote the diagonal elements of the error covariance matrices; the larger the value of $\eta$, the more the uncertainty of $CO_2$ emissions is reduced from the prior state due to assimilation of satellite data. Finally, we computed a metric for the overall correction of flux errors to account for the reduction of bias and random error in the flux estimates.

## 3. Results

### 3.1. Simulation of Satellite Sampling over Cities

We simulated synthetic TanSat-2 sampling over the cities of BJ, JN, LA, and PR on four arbitrary clear-sky days in January and April 2017 (Figure 3). The scattered distribution of cloud-free samples is due to the randomness of small-scale clouds, which are not resolved by the ERA5 cloud data. The differences in $CO_2$ enhancements across the four cities (Figure 4a) are due to different emission levels and footprints associated with local meteorological conditions. The synthetic column $CO_2$ enhancements (with measurement errors) in the four cities range from $-2$ ppm to 8 ppm, similar to previous studies [20,48,74,75]. The mean value of $X_{CO_2}$ enhancements in BJ is 1.19 ppm, with a random error of 0.42 ppm (Figure 4b), resulting in a maximum signal-to-noise ratio of 2.8, followed by JN (2.1), LA (0.7), and PR (0.1). The signal-to-noise ratio of satellite data is an important indicator of flux error reduction, which shows significant correction in BJ and JN and negligible error reduction in PR (Figure 5). The sizable error reduction in each city is concentrated in the region with significant $X_{CO_2}$ enhancement signals and high-density satellite sampling (Figure 3).

### 3.2. Comparison of Flux Inversion for Different Cities

Figures S1–S4 show the spatial distributions of the true, a priori, and a posteriori $CO_2$ emissions and the associated flux noise in the cities of BJ, JN, LA, and PR. The a posteriori emissions optimized by the satellite data can retrieve the true integrated emissions for the four cities within 3% (BJ), 20% (JN), 31% (LA), and 30% (PR) from an a priori state of 13% (BJ), 65% (JN), 52% (LA), and 69% (PR) larger than the truth. The corresponding reductions in flux random errors are 28% (BJ), 19% (JN), 25% (LA), and 23% (PR). The reduced posterior flux noise due to flux correction illustrates that cloud-free satellite data can broadly retrieve the spatial structure and magnitude of the true emissions.

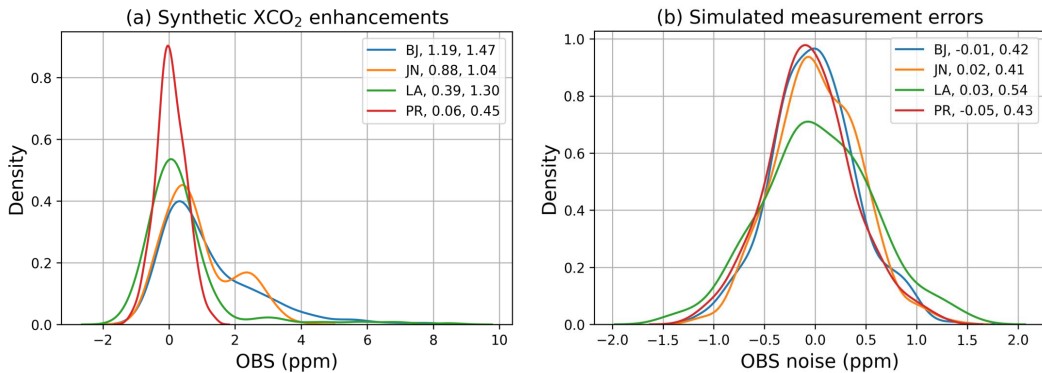

**Figure 4.** Probability density of synthetic $X_{CO_2}$ enhancements in Beijing (BJ), Jinan (JN), Los Angeles (LA), and Paris (PR) (**a**) along with the corresponding measurement errors (**b**). The values (in units of ppm) after the city names are the mean (first number) and standard deviation (second number).

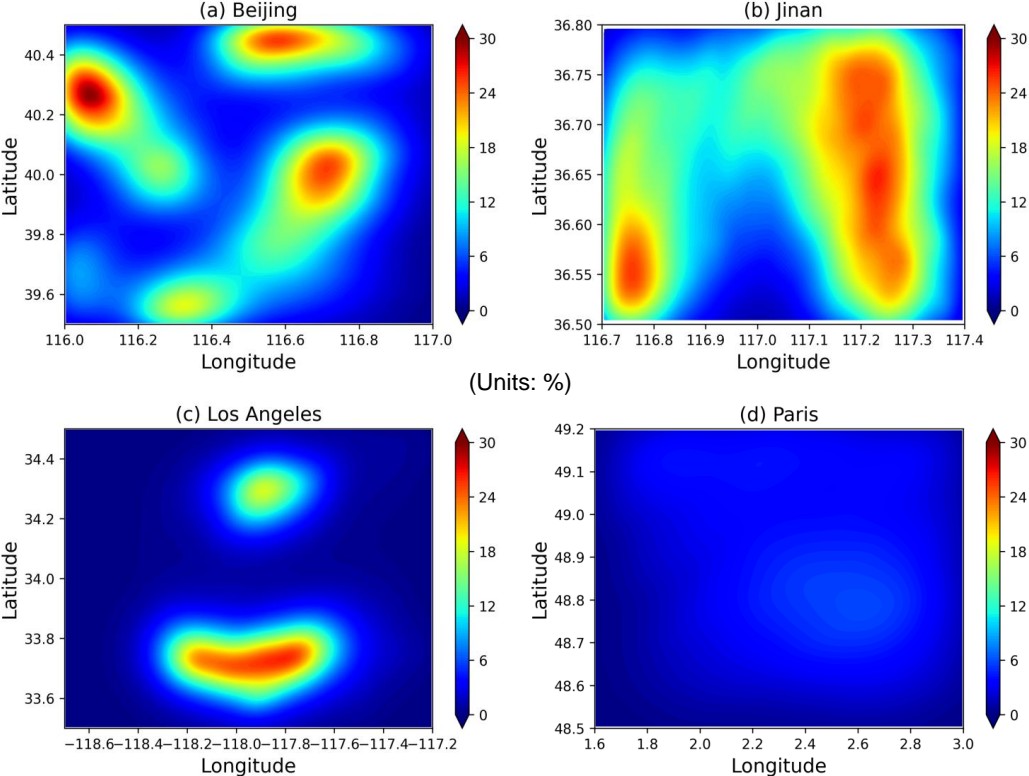

**Figure 5.** Flux error reduction in Beijing (**a**), Jinan (**b**), Los Angeles (**c**), and Paris (**d**).

Figure 6 shows the urban $CO_2$ column enhancements that correspond to the true, a priori, and a posteriori emissions. Certain satellite samples show a significant increase in $CO_2$ (greater than $3\sigma$) in BJ and LA, corresponding to a sizable reduction in flux errors in the two cities (Figure 5). The synthetic $CO_2$ enhancements (synthetic OBS) in PR vary outside the range of truth (perfect OBS) due to the relatively low signal-to-noise ratio. Therefore, the reduction in flux error in PR is negligible (Figure 5). We evaluated the performance of inversion by comparing integrated urban $CO_2$ emissions in the four cities (Figure 7). The bias correction ranges from 40% to 75%, with a 19% to 28% reduction in random flux error (Table 1). The overall correction (including bias and random error) ranges from 32% to 46% depending on the signal-to-noise ratio and the density of satellite sampling (due to clouds and aerosol loading) that can detect significant $X_{CO_2}$ enhancement signals.

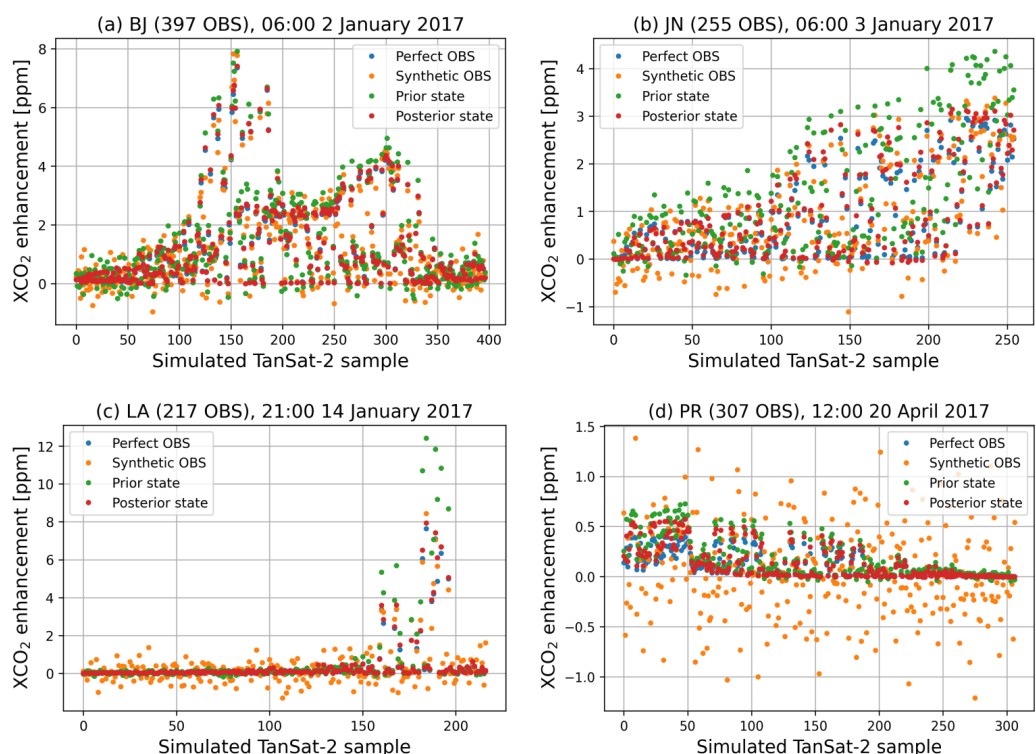

**Figure 6.** Urban $CO_2$ enhancements in Beijing (**a**), Jinan (**b**), Los Angeles (**c**), and Paris (**d**).

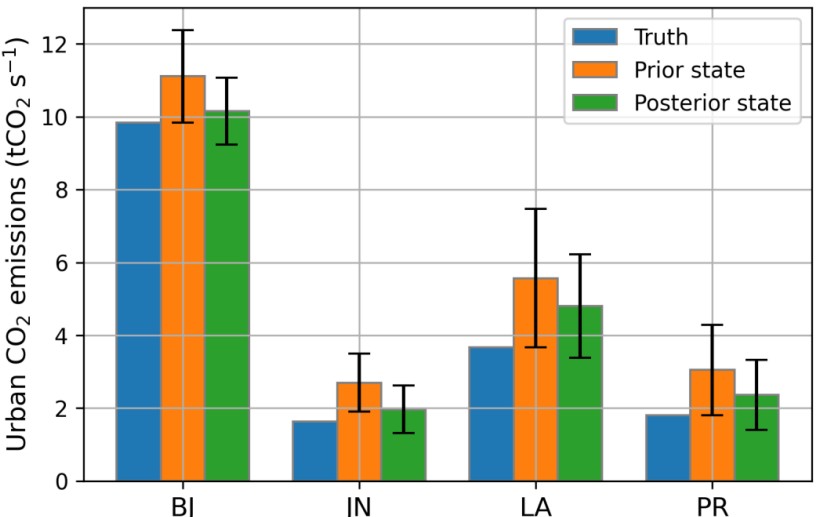

**Figure 7.** Urban $CO_2$ emissions in Beijing (BJ), Jinan (JN), Los Angeles (LA), and Paris (PR).

**Table 1.** Reduction of flux bias and random error (RE) and overall correction (OC) of integrated urban $CO_2$ emissions in Beijing (BJ), Jinan (JN), Los Angeles (LA), and Paris (PR); units are %.

| City | Bias | RE | OC |
|------|------|-----|-----|
| BJ | 75 | 28 | 46 |
| JN | 68 | 19 | 45 |
| LA | 40 | 25 | 32 |
| PR | 56 | 23 | 37 |

### 3.3. Sensitivity to Systematic and Random Measurement Errors

We investigated the sensitivity of flux estimates to bias in satellite data for the medium-size city of JN (Figure 8a), which is a typical city in terms of the area of the city. A systematic measurement error of $\pm 1$ ppm would significantly degrade emission estimates inferred from the data, especially when the observation bias and the flux bias are in the same direction. A slight bias within $-0.5$ ppm in the data, in the opposite direction of a priori flux bias, is beneficial for improving the integrated estimate of $CO_2$ emissions. Figure 8b shows the sensitivity of the flux error reduction to observation uncertainties, including atmospheric transport errors, under different resolutions of satellite sampling. The peak spatially-averaged error reduction is approximately 25% with an observation uncertainty of 0.5 ppm and the largest number of measurements. The lowest error reduction is for the minimum data availability (25 OBS) and a sampling resolution of 20 km. Moreover, the scenario of 4 km resolution (127 OBS) with 1 ppm observation uncertainty shows similar error reduction to the scenario of 8 km resolution (76 OBS), with a 0.75 ppm random error, indicating that better measurement precision can partially compensate for fewer measurements.

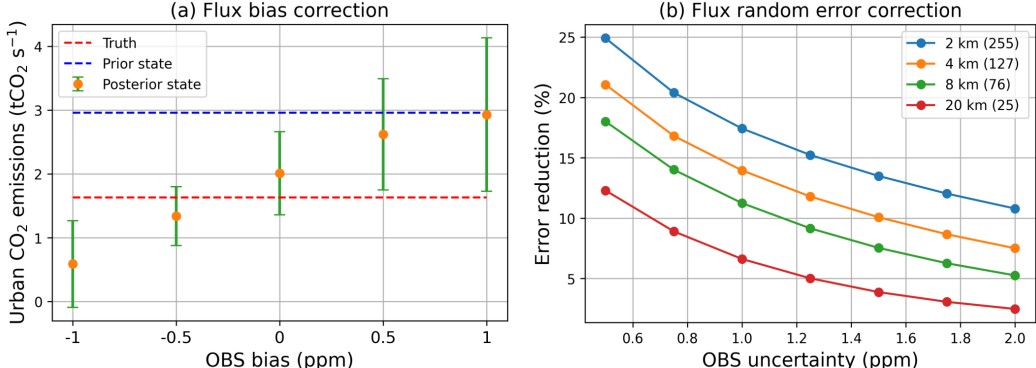

**Figure 8.** Correction of flux bias with the change in observation bias in Jinan (**a**) and reduction of the spatially-averaged random flux error with the change in observation uncertainty under different sampling resolutions in Jinan (**b**). The numbers in parentheses are the number of observations.

## 4. Conclusions and Discussion

In this paper, we demonstrate that it is feasible to infer urban $CO_2$ emissions from synthetic unbiased TanSat-2 data with a 19–28% correction for random flux errors. Because atmospheric column enhancements from urban $CO_2$ emissions are less than 1% of the background concentration, the signal-to-noise ratio of satellite samples is important for determining error reduction in flux estimates. A systematic measurement error of $\pm 1$ ppm would significantly degrade emission estimates derived from the data. Improving the accuracy and precision of satellite samples can help to reduce uncertainties in urban $CO_2$ emissions.

Although we evaluated the potential of using TanSat-2 data to detect anthropogenic $CO_2$ emissions, there are additional limitations and uncertainties in applying these results to inversion experiments using real data [76]. As shown above, the reduction of flux errors is highly dependent on unbiased high-precision (less than 1 ppm random error) satellite measurements. However, satellite measurements are inevitably subject to bias due to the influence of clouds and aerosols. Data collected from ground-based remote sensing instruments (TCCON or EM27/SUN) are valuable to identify and correct larger-scale systematic errors. Deploying ground-based atmospheric remote sensing networks would help to correct regional systematic errors, while random measurement errors can be reduced by increasing the number of individual samples.

Column-averaged urban $CO_2$ enhancements are typically less than 1% of the atmospheric background concentration (about 415 ppm). It has been suggested that estimating the regional background $CO_2$ column concentration is important for quantifying the magnitude of urban $CO_2$ enhancements due to net urban emissions, including anthropogenic

and biospheric $CO_2$ fluxes [77]. A range of methods have been investigated in different studies to quantify regional background values, such as calculating spatial $CO_2$ gradients between upwind and downwind sites [12], solving for concentrations at the boundary as an additional unknown [78], averaging $X_{CO_2}$ measurements over a latitude [79] or over surrounding areas that are relatively unaffected by urban emissions [20], simulating the background concentration with an atmospheric transport model [60], or deriving it from a two-step linear regression [26]. The choice of method for determining the background value depends on the specifics of each study, as all involve simplifications that affect the estimated urban emissions. Although the closed-loop experiment sidesteps this issue, it is necessary to consider additional uncertainties (assessment of wind direction and $CO_2$ uptake by local ecosystems) associated with the calculation of the elevated $X_{CO_2}$ in experiments with real data.

This study assumes unbiased and uncorrelated atmospheric transport errors. However, these errors are likely to be correlated at the sub-city scale [80]. Our assumptions likely result in the best-case scenario for error reduction that can be achieved by the TanSat-2 mission. Better understanding of atmospheric transport and a priori flux errors is essential to improving the accuracy and precision of a posteriori $CO_2$ flux estimates. In addition, seasonal biospheric uptake of $CO_2$ within and around cities weakens observed $CO_2$ gradients, complicating the categorization of anthropogenic and natural fluxes [17,81–83]. Coupled assimilation of $CO_2$ with other trace gases such as CO or $NO_2$ can provide constraints, allowing fossil fuel emissions of $CO_2$ in cities to be separated from natural fluxes [29,32].

**Supplementary Materials:** The following supporting information can be downloaded at: https://www.mdpi.com/article/10.3390/rs15204904/s1, Figure S1: Truth (a), prior state (b), posterior state (c), prior flux noise (prior state minus truth) (d), posterior flux noise (posterior state minus truth) (e), and flux correction (posterior minus prior state) (f) based on the cloud-free samples in Beijing at 06:00 (UTC) 02 JAN 2017. Values in parentheses are the total $CO_2$ emissions within the domain and their uncertainty.; Figure S2: Same as Figure S1, but in Jinan at 06:00 (UTC) 03 JAN 2017; Figure S3: Same as Figure S1, but in Los Angeles at 21:00 (UTC) 14 JAN 2017; Figure S4: Same as Figure S1, but in Paris at 12:00 (UTC) 20 APR 2017.

**Author Contributions:** Conceptualization, K.W., D.Y. and Y.L.; Data curation, D.Y.; Formal analysis, K.W. and D.Y.; Funding acquisition, D.Y. and Y.L.; Methodology, K.W. and D.Y.; Visualization, K.W.; Writing—original draft, K.W. and D.Y.; Writing—review and editing, K.W., D.Y., Y.L., Z.C., M.Z., L.F. and P.I.P. All authors have read and agreed to the published version of the manuscript.

**Funding:** This work was supported by the National Key Research and Development Plan (2021YFB3901000) and the CAS Project for Young Scientists in Basic Research (Grant No.YSBR-037).

**Data Availability Statement:** The ODIAC emission inventory is available at http://dx.doi.org/10.17595/20170411.001 (accessed on 25 September 2023), hosted by the Center for Global Environmental Research, National Institute for Environmental Studies (https://db.cger.nies.go.jp/dataset/ODIAC/, accessed on 7 August 2023). The X-STILT model is available at https://github.com/uataq/X-STILT, accessed on 7 August 2023. Codes for this study are available upon request.

**Acknowledgments:** We thank Dien Wu at CIT for discussing the X-STILT transport model and Tom Oda at USRA for publishing the ODIAC emissions inventory.

**Conflicts of Interest:** The authors declare no conflicts of interest.

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
