# Peer review of "Evaluating the Ability of the Pre-Launch TanSat-2 Satellite to Quantify Urban CO2 Emissions"

_remotesensing, doi:10.3390/rs15204904_

Round 1

Reviewer 1 Report

Wu et al. present an observing system simulation experiment study and try to assess how well TanSat-2-based XCO2 could be used to constrain urban emissions with an X-STILT inversion scheme, using four large cities as case studies.

The study itself is well written and the text contains the appropriate level of detail to understand the methods and possibly attempt to recreate the results. The authors used a set of established tools and methods and applied them on a future planned mission, so the novelty factor of the study is somewhat limited. The he study would benefit greatly if the section on the TanSat-2 mission would contain references, otherwise it would appear that this manuscript is the first time this mission is ever mentioned in any detail. It would also make the paper much more interesting to readers.

The bottom line seems to be that in a somewhat idealized scenario absent of realistic transport errors and a perfectly known background, the inversion method can recover the true emission, as long as retrieval biases are less than ~0.5 ppm. That result in itself is rather interesting, as many studies on city emission inversions do not take retrieval biases into account. This also makes the methodology an interesting tool to develop mission-level requirements if a mission goal is to infer urban emissions.

I can recommend the article for publication, and below noted a few suggestions to improve the article.

Line 78: TanSat-2 will measure absorption spectra, not XCO2/XCH4, and those spectral will be turned into XCO2/XCH4 via retrievals and are then used to infer emissions

Line 81: push bloom -> push broom

Line 125: I suggest writing these XCO2 ranges as "-2 ppm to +2 ppm", rather than "-2--2 ppm"

Line 224: Also consider reading this publication by A. Schuh on the impact of biogenic background on fossil fuel emission inversion: https://doi.org/10.1016/j.rse.2021.112473

Caption of Figure 4: the distributions don't show synthetic CO2, but rather the XCO2 enhancements, if I understood the figure correctly

Figure 5: I find this figure very hard to interpret, are those spatial patterns indicative of something? That aspect is not well explained, and makes the appearance of these figures confusing.

Figure 5, S1, S2, S3, S4: I would strongly suggest to never use the rainbow color scheme in figures. Especially when values from -X to +X are used (S1,S2,S3,S4) - there are dedicated diverging colormaps that are much better suited and do not produce artificial gradients in the figures.

Figure 8: I would suggest to re-order the x axis of the left panel to go [-1, -0.5, 0, 0.5, 1].

Author Response

Attached is our response

Reviewer 2 Report

This study investigates the potential of the TanSat-2 satellite (planned to be launched in 2025) to estimate urban CO2 emissions. The manuscript is valuable for understanding the capabilities of the upcoming TanSat-2 satellite. I have only a few minor points and one main concern. I consider the manuscript acceptable after addressing these points.

Main concern:

As the authors mentioned in Line 221-227, column-averaged urban CO2 enhancements are less than 1% of the background concentration. How to reduce uncertainties in quantifying the atmospheric background values is critical for applying these OSSE results to real-data inversions. Please elaborate more details of other previous studies on quantifying regional background values.

Minor points:

Line 30-31: Please specify the range of temporal and spatial scales.

Line 33: “quantify CO2 fluxes at the surface” -> “quantify urban CO2 fluxes”

Line 113: Add references for citing the GFS0.25 data.

Line 135: Specify details of the Chi-square test and add the corresponding references.

Line 176: The four cities selected by this study are representative for different types of cities around the world, but the summary of the comparison of their inversion results is not sufficient, please elaborate the comparison of inversion results in these cities and how to link them to other cities around the world.

Line 195 and Table 1: Specify the definition of the overall correction of flux errors.

Line 211: Please add one or two paragraphs to discuss the impacts of biogenic CO2 fluxes and cloud cover on estimating anthropogenic CO2 emissions in urban areas.

Author Response

attached is our response

Reviewer 3 Report

The manuscript explores the feasibility and effectiveness of TanSat2-satellite in quantifying XCO2 emissions, and analyzes the random and systematic flux errors in each of the 4 case studies, which were conducted in 4 cities respectively. Despite the creativeness of this study, as well as the satellite-based monitoring nature (which is of certain scientific novelty), there are some places that the authors should further elaborate, or add in further details accordingly:

(1) Line 21: For "XCO2 retrieval errors", does it refer to "satellite-based errors, instrumental errors, or resolving capability of monitoring network / instruments"?

(2) Lines 23-24: "The reduction in the systematic flux error for the four cities is sizable, subject to unbiased satellite samples and ..." - does temporal averaging / temporal or spatial discontinuity affect error in systematic flux derivation?

(3) Lines 36-43: The authors should not only focus on the scientific aspects of monitoring tropospheric NO2 or combustion of fossil fuels, but also connect the fossil fuel CO2 emission with carbon reduction policies. Some references are provided as follows:

https://www.sciencedirect.com/science/article/pii/S1462901118312905

https://www.nature.com/articles/s41558-021-01001-0

https://www.sciencedirect.com/science/article/abs/pii/S0269749121016754

(4) Lines 43-46: There are instruments and datasets obtained from different missions and satellites, do the authors mean interpolation of these datasets / combining all these datasets for running model outputs?

(5) Lines 55-56: high-frequency data near metropolitan areas to better constrain the trend of urban CO2 emissions - for "constrain", does it mean understand, or have deeper implication?

(6) Lines 59-60: "The precision of XCO2 measurements is designed to be less than 1 ppm" - how? The error is almost impossible to set it as "less than 1 ppm", please explain how could this be conducted, unless the spatial resolution is extremely high.

(7) Section 2.1: Please state the period of study, especially for the 4 case studies within the 4 countries / cities. Please also state any atmospheric / environmental conditions required.

(8) Line 117: What do you mean by "air parcels arriving at the locations of each sample" - how is the sampling conducted?

(9) Line 130: "formal estimation" - through what manner? Please state the methodology.

(10) Line 148: "Minimizing the cost function" - what is said to be the "cost function"?

(11) Line 175: "minimum reduction of flux error in PR" - why? Please analyze the trend via geographical manner.

(12) Line 185: "from different wind directions" - please show the wind field patterns (figures).

(13) Line 196: How could the correction of "observation density of satellite sampling" be conducted?

(14) Lines 224-227: The authors mentioned that different methods have been investigated in different studies to quantify regional background level, please add in more elaborations. Also, the authors mentioned that "additional uncertainty" has to be considered - what are the additional uncertainties induced?

(15) Line 229: What does it mean by "correlated at the sub-city scale"?

(16) Figure 1 is rather too detailed, should be much clearer in terms of flow diagram (i.e., arrows induced)

(17) Figure 5(d): Please explain more why the flux error reduction is negligible in Paris. What geographical features constitute this?

(18) The authors have provided numerical figures in Table 1. However, some in-text descriptions should also be added in the main text.

(19) Figure S4: Again, please explain why the flux correction in Paris is almost negligible, as compared with other 3 cities investigated.

(20) A proper conclusion and future insights should be added to this manuscript. Currently, it was missing.

The quality of English is mostly good, except some minor places, for example,

Line 159: Add a "comma" after "eta" and before "the"

Line 214: As we "showed" above

Line 222: It is "suggested"

Line 231: Better understanding "of" atmospheric transport

Line 233: complicate the "categorization" of anthropogenic

Author Response

attached is our response

Round 2

Reviewer 3 Report

The authors have improved the quality of the manuscript. There are just some minor modifications needed.

(1) Regarding the reduction in the systematic flux error, the authors should add a short remark somewhere in the manuscript to explain the sources etc.

(2) About the precision of XCO2 measurements being expected to be less than 1 ppm, the authors have not provided a numerical approach to calculate the precision of XCO2 measurements. Please quantify "1 ppm".

Other than that, I believe the manuscript is having good quality to be published.

The English is fine, please just check for any typos (as appropriate)
